# Crosslinking of fibrous hydrogels

Daniël C. Schoenmakers [1], Alan E. Rowan[1,2] & Paul H.J. Kouwer [1]

In contrast to most synthetic hydrogels, biological gels are made of fibrous networks. This architecture gives rise to unique properties, like low concentration, high porosity gels with a high mechanical responsiveness as a result of strain-stiffening. Here, we used a synthetic polymer model system, based on polyisocyanides, that we crosslinked selectively inside the bundles. This approach allows us to lock in the fibrous network present at the crosslinking conditions. At minimum crosslink densities, we are able to freeze in the architecture, as well as the associated mechanical properties. Rheology and X-ray scattering experiments show that we able to accurately tailor network mechanics, not by changing the gel composition or architecture, but rather by tuning its (thermal) history. Selective crosslinking is a crucial step in making biomimetic networks with a controlled architecture.

[1] Institute for Molecules and Materials, Radboud University, Heyendaalseweg 135, 6525 AJ Nijmegen, The Netherlands. [2] Australian Institute for Bioengineering and Nanotechnology, The University of Queensland, Brisbane, QLD 4072, Australia. Correspondence and requests for materials should be addressed to A.E.R. (email: alan.rowan@uq.edu.au) or to P.H.J.K. (email: p.kouwer@science.ru.nl)

Life is supported by hydrogels. They give mechanical properties to cells and their surrounding matrix[1,2]. Nature is able to precisely regulate the stiffness of these gels in space and time. For instance, the most abundant cytoskeletal protein, F-actin is (reversibly) bundled and crosslinked by various actin-binding proteins, resulting in a soft, porous and fibrous network structure[3,4]. It is this bundled network architecture that determines the unique mechanical properties of the network: its stiffness, and also its strong increase in stiffness upon deformation of the gel, the so-called strain-stiffening behavior[2,5]. The linear and nonlinear mechanics are crucial parameters in cellular functions, intracellular communication and tissue protection[6,7]. Other biogels, such as fibrin, collagen, and intermediate filaments show analogous architectures and properties[8,9].

In contrast, synthetic hydrogels that are studied for many biomedical applications commonly have very different architectures (high concentration and dense, single-chain networks) and are not strain-responsive[6]. To manipulate mechanics in these networks, one routinely changes the polymer or crosslinker concentration, which simultaneously changes the architecture and, for instance, the pore size, and the density and distribution of (bio)functional groups that are conjugated to the polymer[10,11]. Developing methods to reliably decouple the mechanical properties is still an outstanding challenge.

In this manuscript, we demonstrate a new approach to synthesize networks with different mechanical properties, but all with very similar architectures. The only way to achieve this is to use Nature's approach. In analogy to actin networks, our synthetic hydrogels are soft, porous fibrillar structures[12,13]. We then crosslink the polymers selectively inside the bundles, which keeps the network architecture unchanged (Fig. 1a). By changing the crosslinking conditions, the concentration and the nature of the crosslinkers, we can accurately tailor the mechanical properties, both in the linear and in the strain-stiffening regime. This approach is generic for any bundled hydrogel material, either synthetic or biological.

## Results

**Materials**. As a model system, we employed the semi-flexible (ethylene glycol)-decorated polyisocyanide (PIC) polymers.[13] An aqueous PIC solution reversibly gels when heated above its lower critical solution temperature (LCST) and a branched, bundled gel is formed with an architecture that is broadly distributed in length scales, both in bundle diameters and in pore dimensions. The gel is unique, since it combines the advantages of a synthetic polymer with a strongly biomimetic character in both architecture and mechanical properties[12,14]. Similar to any other gels, its stiffness strongly depends on concentration, polymer characteristics and conditions, like temperature[14] and the ionic strength[15] of the solution. In addition, PIC share the strain-stiffening characteristics with bundled structural biological protein-based gels, which means that the gel can become 10–100 times stiffer on deformation[2,4]. Such bundled hydrogels have great potential in 3D tissue engineering, and also for PIC polymers and gels regenerative medicine[16], immunology[17,18], and (DNA) sensing applications have been reported[19,20].

For crosslinking, we used the efficient strain-promoted azide alkyne cycloaddition (SPAAC) reaction[21,22]. We decorated the PIC polymer with azide-functional groups by co-polymerizing an azide-functionalized monomer (3.3 mol%) with an inert monomer, resulting in PIC polymer **1** (Fig. 1b). Projected along the backbone, the average azide spacing amounts ~3 nm, but because of the helical backbone conformation, adjacent azide groups, however will be spread out much further. We designed two difunctional crosslinkers based on cyclooctynes with spacer lengths

of ~3 nm (**2a**) and ~10 nm (**2b**). The spacer length of the crosslinker is crucial in the selectivity of the crosslinking process. Details of the synthesis and characterization are provided in Supplementary Methods.

To form the crosslinked gel, a pre-cooled aqueous solution of crosslinker **2** is added at once to a similar volume of cold solution of polymer **1**. This solution then is quickly heated into the gel phase where the SPAAC reaction is allowed to proceed for one hour. CryoSEM images of the crosslinked hydrogel **1 + 2a** (Fig. 1c) show a porous network with features similar to the corresponding hydrogel without the crosslinker present (Fig. 1d). At low temperatures, however, the network structure of crosslinked gel is conserved (Fig. 1e), while the corresponding SEM image of the non-crosslinked material shows an unstructured polymer film (Fig. 1f).

**Mechanical properties of crosslinked bundled networks**. Rheology experiments (Fig. 2) confirm this behavior. In a typical experiment, the crosslinker is added to a cold PIC polymer solution and the mixture is transferred to the rheometer, which measures the storage modulus $G'$ as the sample heats up (Fig. 2a). Both samples with (**1 + 2a**, blue line) and without (**1**, orange line) crosslinker show the same clear transition from a low viscous solution with minimal modulus to an elastic gel (LCST = $T_{gel} \approx$ 15 °C). In the gel phase, the bundles are held together by non-covalent interactions, which despite their weak character barely dissociate. Even under stress, network relaxation or creep is minimal. The addition of a small amount of crosslinks does not impact the mechanical properties of the material in the gel phase, *i.e.* above $T_{gel}$. The effect of the crosslinking reaction does become very clear when the samples are cooled again: whilst the sample without crosslinks displays the reverse gelation transition back to the polymer solution (dashed orange line), the modulus of the crosslinked hydrogel stays high and only marginally decreases to $G' \approx 50$ Pa at $T = 5$ °C (dashed blue line). The covalent linkages in the network prevent the hydrogel from disintegrating. The preservation of the bundled network architecture is clearly illustrated by the mechanical response of the gel to high stress. At high stresses, i.e., beyond a critical value, PIC gels, as well as other semi-flexible networks show a stiffening regime, where the differential modulus $K' = \partial\sigma/\partial\gamma$ ($\sigma$ and $\gamma$ are the stress and strain, respectively) scales to the applied stress: $K' \propto \sigma^m$ for $\sigma > \sigma_c$ (where $m$ and $\sigma_c$ are the stiffening index and the critical stress, respectively). The characteristics of this regime strongly depend on the bundle structure[2,5,14,23,24]. In the crosslinked **1 + 2a** hydrogel, the stiffening responses at 5 and 37 °C (Fig. 2b) are virtually identical, which strongly indicates that the load-bearing network in the gel is conserved.

The linear and nonlinear mechanical properties of semi-flexible networks are the direct consequence of the bundled structure. How then can this bundled structure be conserved below the gelation temperature with such low concentration of crosslinkers [**2a**] = 52 μM? Note that a typical crosslinker concentration in a soft polyacrylamide gel is 100 times higher: 2−10 mM. To answer this question, we need to consider the molecular design of the polymer and the crosslinker. The polymer is relatively stiff and ordered in a ~$4_1$ helical conformer. With 3.3% azide content, adjacent azide groups will on average be at least ~5 nm apart. Crosslinker **2a**, however is much shorter than that, which means that intramolecular crosslinking is insignificant. Also, the intermolecular crosslinking rate in solution is very low, because of the high dilution (52 μM) of functional groups in the sample. The only opportunity to generate crosslinks is above $T_{gel}$, when bundles form. This process effectively concentrates the azides and, as a result, the crosslinks are predominantly formed between

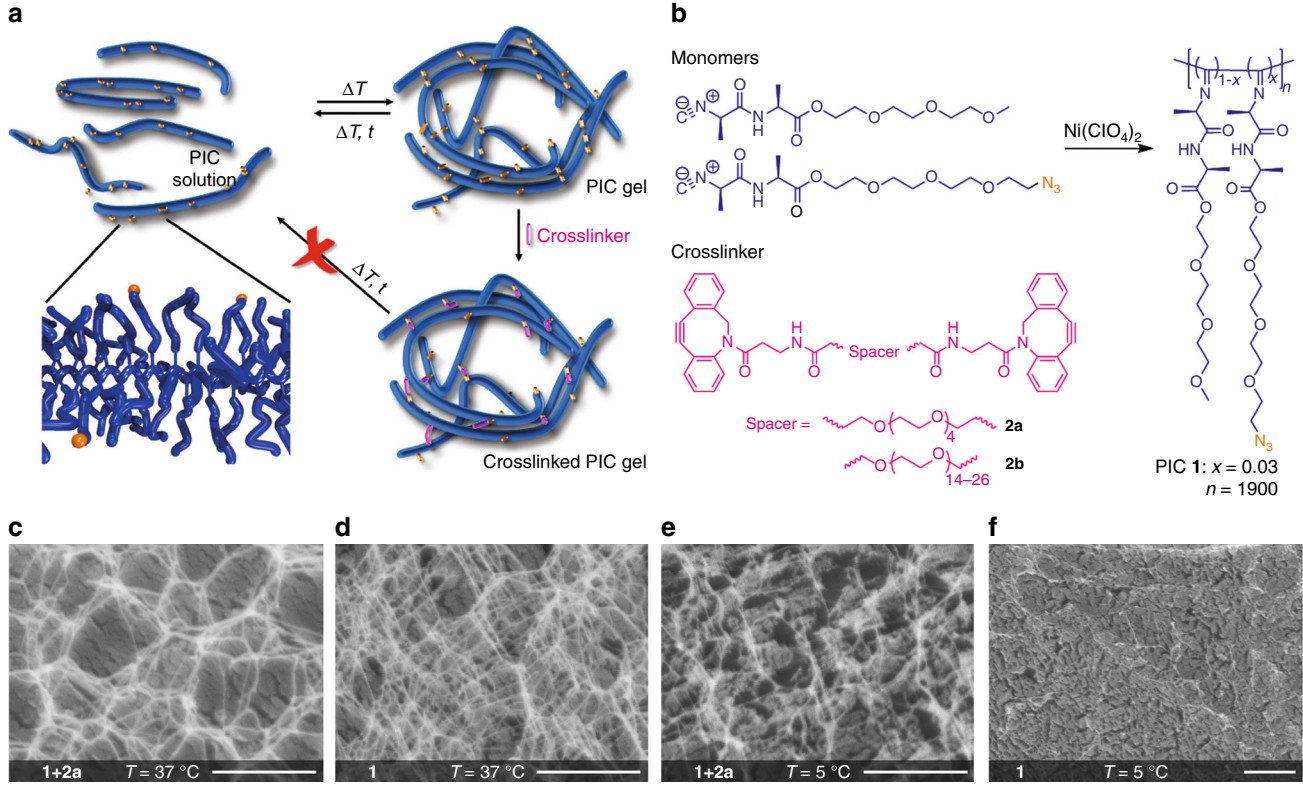

**Fig. 1** Crosslinking a bundled network. **a** Schematic representation of the crosslinking method. Azide (orange) decorated polymers (blue) are gelled and crosslinked selectively within the bundles by a crosslinker (pink), stabilizing the architecture. **b** Gel components: PIC **1** (yield: 94%, 3.3-mol-% $N_3$, $M_v =$ 599 kg mol$^{-1}$) and crosslinkers **2a** and **2b**. **c–f** Freeze-fractured cryoSEM micrographs of PIC gels (scale bars=1 μm): crosslinked (**c**) and not-crosslinked (**d**) gel at $T = 37\,°C$ and crosslinked (**e**) and not-crosslinked (**f**) gel at $T = 5\,°C$. Note that in the final cryoSEM image, the network structure has disappeared

polymer chains inside a bundle, in other words intra-bundle, but intermolecular. On cooling again, this characteristic bundled network architecture is maintained. To verify that efficient crosslinking indeed requires bundle formation, a prolonged crosslinking procedure was performed at temperatures below $T_{gel}$ (5 °C and 18 °C in the presence of 1.0 M NaI). Even after 2 h, we observe no sign of efficient network formation (Fig. 2c, Supplementary Fig. 1). After heating this mixture into the gel phase and subsequently cooling it back to 5 °C, however, an elastic gel is present. The intra-bundle crosslinking approach has the great advantage that crosslinks are very effective and the open porous structure of the gel is remained, which is critical for many biomedical applications. In addition, this crosslinking approach is generic and can be applied to any bundled network, biological or synthetic.

**Controlling mechanics with temperature.** In addition, PIC hydrogels uniquely continue to stiffen beyond $T_{gel}$ because of the increasing stiffness or persistence length of the individual polymer chains following: $G' \sim Te^{2\beta T}$ with thermal coefficient $\beta = 0.055$ K$^{-1}$ for **1** and **1 + 2a** (Fig. 2a), which roughly corresponds to a tenfold increase in the stiffness every 20 °C. The gel structure during this heating process does not change significantly[12], which means that the thermal stiffening behavior is solely attributed to an increase in the persistence length of the bundles. When the crosslinking reaction captures the architecture and the mechanical properties at the crosslinking conditions, varying the crosslinking temperature should give us hydrogels with different mechanical properties, but now with a very similar network architecture; a highly uncommon characteristic of hydrogels and much pursued to study the pure effects of mechanical properties

on cell behavior. Since they show different persistence lengths, the structure inside the bundles likely varies with temperature, although the disordered nature of the bundles (compared to biological gels) hampers experimental verification.

We prepared five gels (**1 + 2a**) that were crosslinked at 25, 30, 40, 50, and 65 °C. After mixing **1** and **2a** at 5 °C and applying the solution to the rheometer, the temperature was raised immediately to the desired crosslinking temperature $T_{cl}$ where bundles form and crosslinking commences (Supplementary Figs. 2 and 3). After 1 h, the crosslinked gels were cooled to 5 °C while measuring the storage modulus (Fig. 2d). Indeed, samples crosslinked at a higher temperature are much stiffer over the entire temperature range than samples crosslinked at lower $T_{cl}$. Looking at 20 °C, for instance, the modulus of the sample crosslinked at 65 °C is about 10-fold higher than that of the sample crosslinked at 25 °C. This approach offers a unique opportunity to truly study mechanics without the influence of other parameters, such as network composition or architecture.

The samples that were polymerized at different temperatures were then heated from 5 °C all to 50 °C (Fig. 2e). Up to the crosslinking temperature, $G'$ follows the same trace as the cooling ramp, showing full reversibility without hysteresis. Beyond $T_{gel}$, however, the slope of $G'$ vs $T$ increases to the value of the pristine heating ramp ($G' \sim Te^{2\beta T}$, with nearly identical $\beta = 0.051$ K$^{-1}$), resulting of convergence of the mechanical properties of all gels at elevated temperatures, irrespective of their crosslinking temperature. Despite the presence of the crosslinks, the polymer chains and with them, the bundles continue to stiffen. In subsequent cooling ramps (Fig. 2f), we find that this stiffening process is irreversible and all samples show similar mechanical properties, independent of the original crosslinking temperature. We

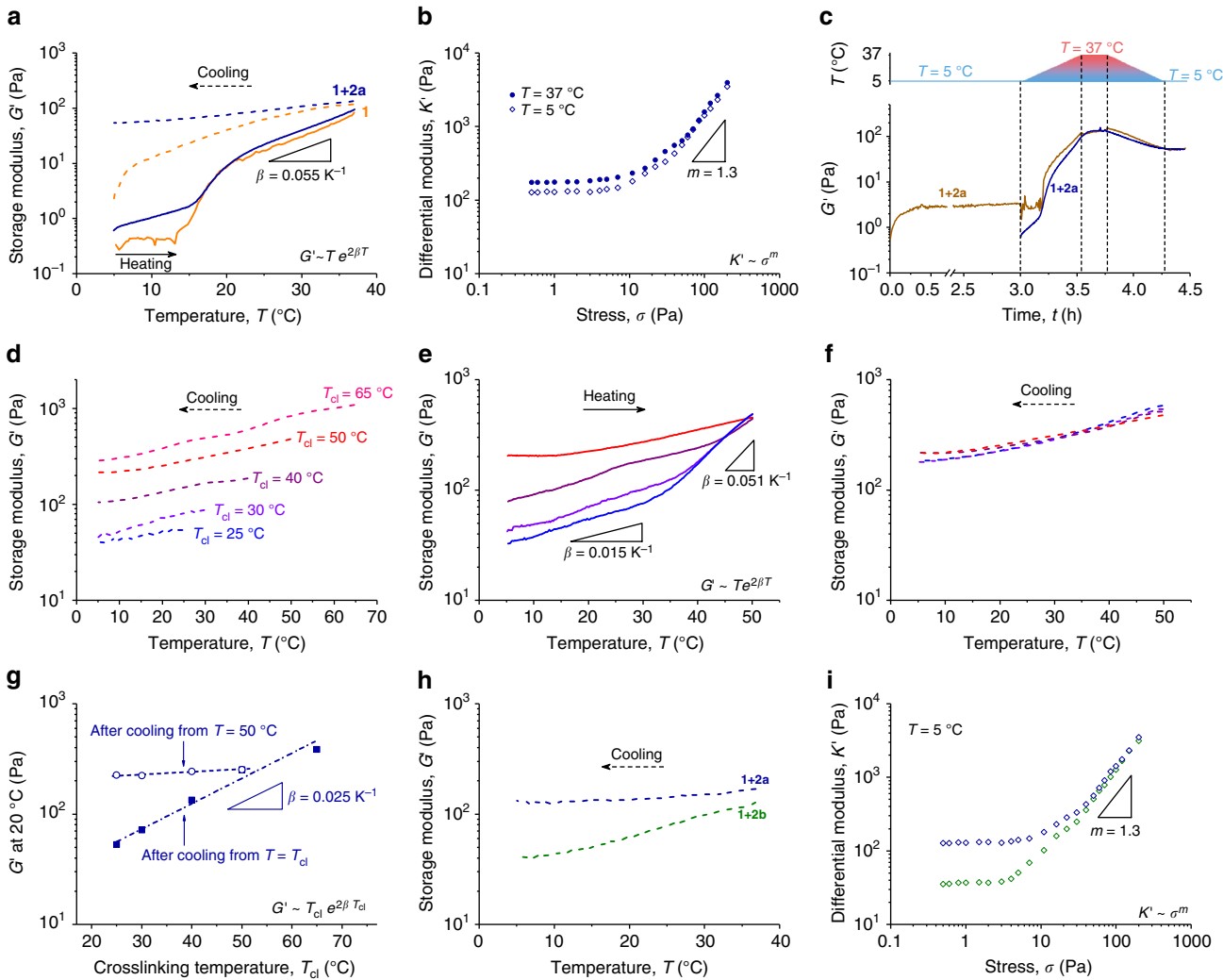

**Fig. 2** Mechanical properties of intra-bundle crosslinked PIC hydrogels. **a** Storage modulus $G'$ of a crosslinked (**1** + **2a**, blue data) and a not-crosslinked (only **1**, orange data) PIC gel. The heating (solid lines) curves shows gel formation at 15–20 °C. After crosslinking at 37 °C, the cooling (dashed lines) curve of **1** shows 'melting', while **1** + **2a** remains a gel. The thermal stiffening exponent $\beta$ originates from single chain stiffening[14]. **b** Strain-stiffening: differential modulus $K' = \partial\sigma/\partial\gamma$ as a function of applied stress $\sigma$ for the **1** + **2a** gel (crosslinked at $T_{cl} = 37$ °C) and measured at 5 °C (open diamonds) and 37 °C (solid circles). Below a critical stress $\sigma_c$, $K' = G'$, above it $K' \sim \sigma^m$, with $m$ the stiffening index, quantifying the responsiveness to stress. **c** Crosslinking in the absence of bundles (at $T_{cl} = 5$ °C, light blue data) does not lead to network formation, but as soon as the temperature is raised and bundles form, the gel forms permanent crosslinks that are stable on cooling, similar to that of the immediately crosslinked sample (blue data). **d** Storage moduli $G'$ of **1** + **2a** gels crosslinked at $T_{cl} = 25$ °C (blue), $T_{cl} = 30$ °C (violet), $T_{cl} = 40$ °C (purple), $T_{cl} = 50$ °C (red) and $T_{cl} = 65$ °C (pink) when cooling from the crosslinking temperature $T_{cl}$ to 5 °C (cooling rate: 1 °C min$^{-1}$). **e** $G'$ of crosslinked PIC gels ($T_{cl} = 25$–50 °C), reheating from 5 °C to 50 °C with a heating rate of 1 °C min$^{-1}$. **f** The moduli of crosslinked PIC gels ($T_{cl} = 25$–50 °C), cooling from 50 °C to 5 °C with a heating rate of 1 °C min$^{-1}$ fully overlap. **g** The moduli of crosslinked gels at $T = 20$ °C, measured after cooling from their respective crosslinking temperature (red data) and after cooling from $T = 50$ °C (black data). **h** Effect of the length of the crosslinker: $G'$ of gels crosslinked with a 'long' crosslinker **2b** (green) and 'short' crosslinker **2a** (blue); both cooling from $T_{cl} = 37$ °C to 5 °C (−1 °C min$^{-1}$). **i** Differential modulus of the hydrogels crosslinked with the 'long' (**2b**, green) and the 'short' (**2a**, blue) spacer, crosslinked at $T_{cl} = 37$ °C and measured at $T = 5$ °C. For all experiments in Fig. 2, the concentrations were identical: [**1**] = 1 mg mL$^{-1}$, equivalent to [N$_3$] = 104 μM; crosslinkers: [**2a**] or [**2b**] = 52 μM (or 0 μM in panel **a**)

anticipate that heating beyond $T_{cl}$ irreversibly tightens and stiffens the bundle which exposes residual reactive groups that crosslink and irreversibly stiffen the hydrogel[19]. When we plot the moduli of the gels at $T = 20$ °C for samples crosslinked at different $T_{cl}$ (Fig. 2g), we observe a clear difference in stiffness when they are directly cooled after crosslinking where $G'$ scales not to the actual temperature, but to the original crosslinking temperature: $G' \sim T_{cl} \exp(2\beta T_{cl})$. Once the samples are heated to 50 °C, their difference in mechanical properties at 20 °C has disappeared.

**Crosslinker design**. By modifying the length of the crosslinker, we were able to further tailor the network structure and the mechanical properties of the hydrogels. As an illustration, we designed and prepared (see Supplementary Methods) crosslinker **2b**, which is much longer (~10 nm) than **2a**, but still an order of magnitude smaller than the pore size. Hence, **2b** is still expected to give predominantly crosslinks inside the bundle, but will less efficient in keeping the polymer chains in a tight bundle below $T_{gel}$. Indeed, although prepared at the same concentrations and conditions ($T_{cl} = 37$ °C, 52 μM), the stiffness of gels crosslinked

with the longer spacer **2b** decreases much stronger with on cooling than gels with **2a** (Fig. 2h). Cooling below $T_{gel}$ reduces the non-covalent interactions between the polymers and while the short crosslinks sustain the tight bundle structure, the longer crosslinks allow for disorganization of the chains in the bundle and consequently, a reduction in the bundle persistence length and the stiffness of the gel. Since the bundled architecture is still in place, the gel remains elastic ($G' \gg G''$) and, also, the nonlinear properties are conserved at 5 °C (Fig. 2i). In fact, we find that $K'$ of **1 + 2b** and **1 + 2a** fully overlap in the nonlinear regime, which is in line with theory[3,25,26] that predicts that compliant cross-linkers have a limited effect on the strain-stiffening behavior of semi-flexible networks.

**Network architecture.** The network architecture plays a crucial role in the mechanical properties of the gel, but is challenging to study in-situ, due to the low polymer concentrations and the large distributions in bundle and pore dimensions. We used small angle X-ray scattering (SAXS) to characterize the network structure[12]. A low-temperature PIC polymer **1** solution containing free polymers ($T = 5$ °C) is comfortably described by a worm-like chain model by Kholodenko[27], which yields the persistence length ($l_{p,0} = 30$ nm) and polymer diameter ($R = 1.0$ nm) from the fitting procedure. In the gel phase, the scattering traces were fitted to the same Kholodenko model (now yielding the average bundle diameter, $R_B$) together with the Ornstein–Zernike model[28]

which describes the porous network structure. As expected, at high temperatures, crosslinked gel **1 + 2a** shows a scattering pattern analogous to the non-crosslinked hydrogel (Supplementary Fig. 4). When the crosslinked sample is cooled to $T = 5$ °C, the scattering pattern becomes a combination of the pattern of the bundled gel with that of a single polymer (Fig. 3a), i.e., a fraction of the network is re-solvated. The scattering curve is well described by a linear combination of both contributions: the Kholodenko/Ornstein–Zernike model for the bundled network plus a single polymer Kholodenko model (Fig. 3b, c), using most of the earlier found fitting parameters (see Supplementary Methods for the full fitting procedure and Supplementary Table 2 for the fitting results).

We then analyzed the architectures of gels that were cross-linked at different temperatures, by recording and fitting their scattering patterns at the crosslinking temperature $T_{cl}$ (Fig. 3d) after cooling (to $T = 5$ °C, Fig. 3e), heating ($T = 50$ °C, Supplementary Fig. 4e) and cooling again ($T = 5$ °C, Supplementary Fig. 4f). For gels crosslinked at $T_{cl} = 25$ and 30 °C, that is close to the gelation temperature $T_{gel}$, we find a scattering contribution of single polymers that are not yet incorporated in the network. At higher $T_{cl}$, this contribution is negligible ($< 2$ %). Similarly, the average bundle radius $R_B$ of the gel crosslinked at 25 °C, is lower than the gels prepared at higher $T$, either crosslinked or not (Fig. 3f). On cooling non-crosslinked samples, the bundles swiftly disappear, but the SAXS results of the crosslinked gels indicate the presence of a fraction of single polymers, as well as bundles

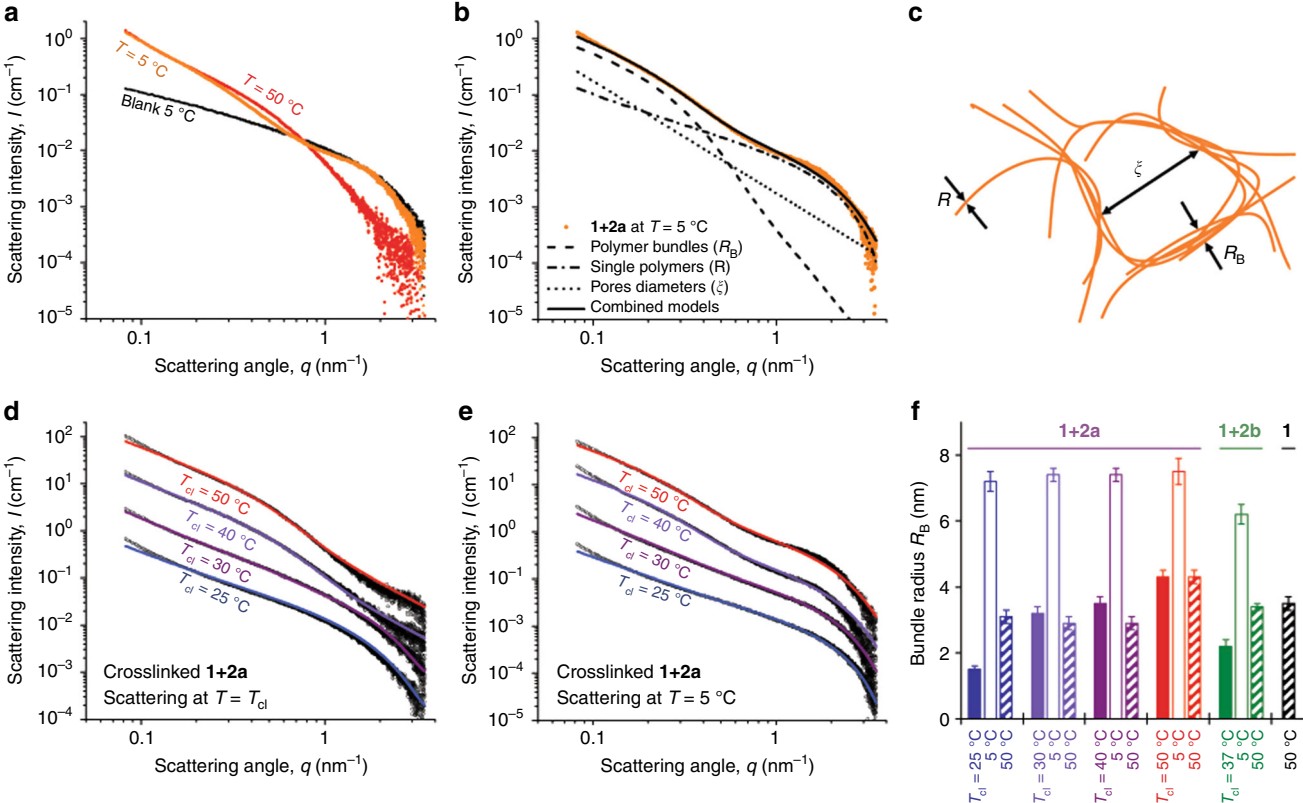

**Fig. 3** Architectural analysis with SAXS. **a** The scattering curve of the **1 + 2a** crosslinked gel at 5 °C (orange) is a combination of the bundled hydrogel pattern (red, **1 + 2a**, 50 °C) and the dissolved polymer pattern (black, **1**, 5 °C). **b, c** Contributions of the models for crosslinked **1 + 2a** gel at 5 °C and model interpretation of architectural length scales. **d, e** SAXS curves of **1 + 2a** gels, crosslinked at 25, 30, 40, and 50 °C at the crosslinking temperatures (**d**) and after cooling to 5 °C (**e**) with the best fit to the model (solid lines). The curves were shifted vertically to enhance clarity; un-shifted curves are given in Supplementary Fig. 5. **f** Average bundle radii of gels crosslinked gels **1 + 2a** (blue to red) and **1 + 2b** (green) at the crosslinking temperature $T_{cl}$, at 5 °C and at 50 °C, as well as the bundle radius of **1** at 50 °C for comparison. All fits include a 40% distribution on the bundle diameter; the error bars are an estimate for the range where the fit yields acceptable results. The concentrations for all samples in the figure are [**1**] = 4 mg mL$^{-1}$ and [**2a**] = [**2b**] = [N$_3$]/2 = 208 μM

with an increased average diameter (Fig. 3b, f). In the gel, bundle diameters are highly distributed (Fig. 1c–e) and thinner bundles will have fewer crosslink points and thus a higher probability to rehydrate and disassemble at low temperatures. Hence, the core of the network at 5 °C is formed by the remaining thicker bundles. These thicker bundles carry the majority of the mechanical load, as implied by the minor reduction in stiffness of the crosslinked gels on cooling and the remaining non-linear response to stress (Fig. 2b). Reheating the gels to 50 °C reduces $R_B$ again to values that are expected for the gel at that temperature. The small bundles formed by reheating do reduce the average bundle size, but seem to play a marginal role in the mechanical properties of the hydrogel. The bundles of **1** + **2a**, crosslinked at 25 °C now also have the expected bundle dimensions and the fraction free polymers has disappeared for all gels. Analogous effects were found for gels of **1** + **2b**. All scattering curves (Supplementary Figs. 4 and 5) and fitting results (Supplementary Table 2) are given in the Supporting Information.

**Crosslink density**. Besides the crosslinking temperature and dimensions of the crosslinker, we examined a third tool to tune the mechanical properties: the crosslink density[29]. We varied the concentration **2a** and, after crosslinking at 37 °C, measured the mechanical properties at 5 °C. Each crosslinker contains two DBCO moieties, so for clarity the mechanical properties were plotted against the number of DBCO moieties per azide group (Fig. 4a). Not unexpectedly, at a DBCO/azide ratio around 1, the gel is stabilized most efficiently, as there the crosslink density will be optimal. UV–vis experiments[30] that afford the conversion of the DBCO moieties confirm that the yield of the crosslinking reaction is then highest (Fig. 4b and Supplementary Fig. 6). Based on the assumption that two reactions need to occur with the same crosslinker to form an active crosslink, we estimated the crosslink density (Fig. 4a), which coincides well with the material's stiffness at 5 °C. At insufficient [**2a**], the crosslink density below $T_{gel}$ is simply too small to form a percolating bundled network that can carry any load; at too high [**2a**], many azide groups are substituted with a crosslinker that is not attached to the network with its other end, which similarly decreases the crosslink density. UV–vis analysis further shows that despite the low concentration, the crosslinking reaction is effectively finished in a few minutes. In addition, at 5 °C the SPAAC reaction proceeds, albeit very slowly, to a conversion of ~50% (Fig. 4b). At this point, the crosslinker reacted with one of its DBCO unit, but the probability of finding a second azide nearby is small and crosslinks are formed sparsely, which is in line with the results shown in Fig. 2c.

**Discussion**

Covalent crosslinking is the default method to stabilize hydrogels permanently, but often results in gels with small pores with limited application potential in 3D cell studies. Biology solved this challenge by generating open porous networks of semi-flexible bundles that are crosslinked by dedicated proteins. We followed this example and presented an approach to crosslink polymers predominantly inside the bundles. Provided that the crosslinking reaction is carried out in the presence of the bundles, crosslinker concentrations of 50 μm are sufficient to stabilize the architecture. For the PIC gels, presented here, bundle formation is thermally induced (and reversible), which means that crosslinking should take place above the gelation temperature, which is readily tuned between 10 and 60 °C by simply changing the ethylene glycol tails[31]. We find that the mechanical properties at the crosslinking conditions are irreversibly captured and cooling shows minor impact on the architecture or the mechanical properties of the material. Further heating, on the other hand will continue to

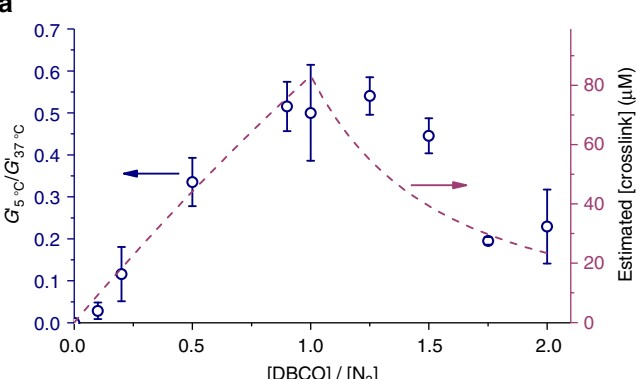

**a**

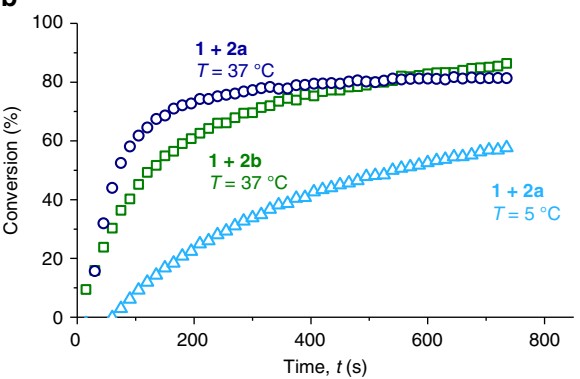

**b**

**Fig. 4** Crosslinker ratios and conversions. **a** Relative stiffness of crosslinked **1** + **2a** hydrogels ($T_{cl}$= 37 °C, [**1**] = 1 mg mL$^{-1}$), after 11 h stabilization at 5 °C as compared to the stiffness of the same gel at 37 °C as a function of the [DBCO]/[N$_3$] ratio (blue data). The error bars represent the standard deviation of $n = 2$ or 3 measurements. The ratio of [DBCO]/[N$_3$] was varied between 0 and 2. Note that the graph shows the relative decrease of the modulus at 5 °C that is normalized to the modulus at 37 °C, because the DBCO units that are present at different concentrations for different crosslinking ratios shift the gel transition temperature slightly, and with that the stiffness of the gel. The absolute values of the moduli at $T$ = 5 °C and 37 °C are given in Supplementary Table 1. Estimated crosslink concentration based on statistical reactions between the crosslinker and the azide and the experimentally determined conversion values (red dotted line). **b** DBCO conversion for gels crosslinked with **2a** or **2b**, at 37 °C and at 5 °C in time measured using UV–vis spectrometry. Colors are explained in the panel

stiffen the network irreversibly. The nature (i.e., length), concentration and, most likely, the functionality of the crosslink are versatile parameters to tailor the mechanothermal response.

This approach is not restricted to polyisocyanide gels, but will be applicable to any bundled gel architecture and will be most effective when the dimensions of the crosslinker are smaller than the distance between the complementary functional groups on the polymer backbone. The genericity renders this strategy highly relevant for biomechanical studies that concentrate on the effect of the tissue mechanics on cell fate.

In addition, our scattering results show that upon cooling a crosslinked gel only the thickest bundles remain, while rheology experiments demonstrate that the stiffness and the mechanical response to stress of the crosslinked gels are barely impacted on cooling. We conclude that in such networks with a large distribution in bundle diameters, the mechanical load is predominantly carried by the thickest bundles and that thinner bundles hardly contribute to the linear or the nonlinear mechanical properties. This conclusion has a significant effect on

how we should visualize stress-development in a polydisperse fibrillar network: what are the length scales of (local) deformation, when for instance a cell adheres to a network and strains it?

## Methods

**Materials**. The synthesis and chemical characterization of **1** and **2b** is described in Supplementary Methods. Crosslinker **2a** is commercially available.

**Sample preparation**. For rheology and UV–vis experiments, a solution of **1** (2 mg mL$^{-1}$) was prepared by overnight dissolving dry polymer in the appropriate amount of milliQ water at 4 °C with occasionally shaking. All polymer solutions were stored at −20 °C prior to use. A solution **2a** (2 mg mL$^{-1}$ in DMSO) or **2b** (2 mg mL$^{-1}$ in milliQ water) was diluted with milliQ water such that an equivo-lumnar mixture of polymer and crosslinker contains the desired ratio azide and DBCO groups (most often 1:1). The solutions were mixed in a pre-cooled glass vial, briefly homogenized and used immediately. For SAXS analysis, more concentrated samples were used: **1** was dissolved in MilliQ water (4.36 mg mL$^{-1}$), as described above and **2a** (4.48 mg mL$^{-1}$) was dissolved in DMSO/milliQ (1:1). For each sample, 16 μL of the **2a** solution and 200 μL of the PIC stock solution were quickly mixed and immediately transferred into a quartz capillary. The capillary was then placed in a water bath at the appropriate $T_{cl}$ for 1 h, after which the scattering experiments were conducted.

**Mechanical characterization**. Rheological measurements were carried out on a TA Instruments Dynamic Hybrid Rheometer 1 or 2 fitted with a 40 mm sand-blasted parallel plate geometry. A geometry gap of 500 μm was used. To measure the linear regime ($G'$), the sample was heated or cooled to the desired temperature, and after a short waiting period for equilibration the complex modulus $G^*$ was measured by applying an oscillating deformation of amplitude $\gamma = 0.04$ at frequency $f = 1$ Hz. Temperature-dependent measurements were carried out at a 1 °C min$^{-1}$ ramp rate. For nonlinear measurements the pre-stress protocol described by Broedersz et al.[5] was used.

**Small angle X-ray scattering**. SAXS data were recorded at the BM26B station at the European Synchrotron Radiation Facilities in Grenoble. Details of the setup are in Supplementary Methods. The recorded curves were fitted using SASfit soft-ware[32]. In brief, single-polymer solutions were fitted to Kholodenko's wormlike chain model[27]:

$$I(q) = I_{\text{polymers}}(q) = (\Delta\rho)^2 \varphi P_0\left(q, L, 2l_p\right) P_{CS}(q, R) \quad (1)$$

where scattering intensity $I(q)$ for every wave vector $q$ is a function of the difference in electron density between the polymer chain and the solution $\Delta\rho$, the polymer volume fraction $\varphi$, and semi-flexible polymer chain characteristics, like the contour length $L$, the persistence length $l_p$ the radius $R$. Scattering profiles of the gel network architectures were described using a combination of the wormlike chain model (now for the bundles) and the Ornstein and Zernike (OZ) model[28] for network heterogeneities:

$$I(q) = I_{\text{bundles}}(q) + I_{\text{network}}(q) \quad (2)$$

$$I(q) = (\Delta\rho)^2 \varphi P_0\left(q, L_B, 2l_{p,B}\right) P_{CS}(q, R_B) + \frac{I(0)}{1 + q^2 \xi_{OZ}^2} \quad (3)$$

Now, $L_B$ and $l_{p,B}$ are the contour length and the persistence length (both beyond the experimental window). The polymer bundle radius $R_B$ contains a normal distribution that represents inhomogeneity of bundle diameters. $I(0)$ is the forward scattering of the OZ-term, and $\xi_{OZ}$ is the correlation length of the network het-erogeneities. For crosslinked gels at low temperatures, a fraction of the gel scatters as single polymers, which was captured by using a linear combination of the two aforementioned models:

$$I(q) = I_{\text{polymers}}(q) + I_{\text{bundles}}(q) + I_{\text{network}}(q) \quad (4)$$

For the analysis, many of the fitting parameters were obtained from control experiments and subsequently fixed. A discussion on the analysis procedure is included in Supplementary Methods.

**Data availability**. All data supporting the results of this study are available in the article and Supplementary Information Files or from the corresponding author on reasonable request.

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

## Acknowledgements

We thank Geert-Jan Janssen (RU General Instrumentation lab) and Dr. Rob Mesman (RU Microbiology) for assistance with the cryo-SEM microscopy and Prof. Giuseppe Portale (University of Groningen) for fruitful discussions on the SAXS analysis. We acknowledge the European Synchrotron Radiation Facilities (ESRF) in Grenoble, France and their staff, in particular Dr. Daniel Hermida Merino, as well as NWO for providing and supporting beam time at the Dutch-Belgium beamline (DUBBLE) for SAXS experiments (Grant BM26-02773). This work was financially supported by NWO Gravitation (Grant 024.001.035).

## Author contributions

D.C.S. and P.H.J.K. designed and interpreted the mechanical and architectural studies. A. E.R. and P.H.J.K. supervised the project. All authors contributed to the manuscript.

## Additional information

**Competing interests:** The authors declare no competing interests.

