## [Peer Review File · Nature Communications]

Reviewers' comments:

Reviewer #1 (Remarks to the Author):

The authors create hydrogels consisting of bundled polyisocyanide (PIC) polymers by heating a solution above its LCST. The resulting bundled hydrogels are then fixated using difunctional crosslinkers connecting two azide groups along the polymer backbone. For the crosslinker, two different lengths are used; 3nm (roughly the inter-azide spacing distance along the PIC backbone) and 10 nm.

The manuscripts argues that this chemistry provides control over mechanical properties in three ways:

- Crosslinking temperature
- Crosslinker design
- Crosslink density

Each of these three is quite convincingly demonstrated to have an effect on the mechanics.

I find these results interesting, novel, and sound. There is certainly sufficient evidence that each of the three control methods has some effect, but my main objection is that the manuscript does little more than note these effects. I find the mechanistic understanding of what is happening lacking, and would have very much liked to see the authors make more of an effort to present a coherent scenario for the changes to polymers, bundles, and network their interventions provoke, and how these lead to the observed changes in mechanical response.

The crosslinking scenario itself is clear enough; above LCST polymers are closer together in bundles giving particularly the shorter linker much more opportunity to bind intramolecularly. Much of these links survive cooling down again, preserving - to some extent - the network structure.

Two claims in the manuscript in particular deserve further motivation or explanation.

1) What does it mean that 'crosslinking captures the architecture and the mechanics'? One of the principal reasons that networks stiffen is that not only the polymers, but particularly the bundles behave as stiff (or as these authors say semiflexible) chains. Surely, the additional intrabundle crosslinks change this effective stiffness profoundly in a largely non-temperature related way, so why would they approach a common asymptote? Clearly it cannot both be true that architecture and stiffness are preserved, because then the low-temperature parts would be the same too, right?

2) Why is the plateau modulus for the 2b linker so much lower? Ascribing it to the 'loose bundle structure' is insufficient. The plateau modulus depends, typically, on the persistence length, the concentration, and the mesh size - which of these is affected by the longer spacer and how? How does this compare to the bundle radius measurements in Fig 3f? These appear to show a relatively small effect? Could the Kholodenko model not also be used to measure a tentative change in persistence length?

3) The architecture may well be conserved, but I suspect the crosslinking induces and possibly locks in stresses in the gel. These stresses may to offset the mechanical response in ways not considered here at all (but potentially responsible for some of the effects, particularly the higher-T stiffening). How have the authors ruled out that this is the case?

Overall, I found the manuscript interesting but somewhat disjointed and enumerative. Potentially, this may be repaired with more mechanistic or model support.

Reviewer #2 (Remarks to the Author):

The manuscript by Schoenmakers et al. describes a hydrogel material system with temperature-dependent crosslinking that is used to tune the mechanical properties. It is shown that by controlling the temperature of crosslinking, the mechanical properties are altered without changing the architecture of the network. It is suggested that an independent control over the shear storage modulus and network architecture is desired for biological applications. SAXS is used to find the bundle diameter of the crosslinked network and it is reported that the architecture of the gel is composed of both polymer bundles and single polymers.

The approach for secondary covalent crosslinking of the hydrogel within the bundles (intra-bundle) and with an insignificant amount of intramolecular crosslinking is laid out well and may serve as an intelligent design scheme for future work. My main issue is with the potential utility of these materials. Upon gelation at elevated temperatures, covalent intrabundle crosslinking is used to "lock-in" the structure, allowing one to cool the system and retain the mechanical properties and structure. In a biological system where T_{application} is already above the gelation temperature, is much benefit to the covalent crosslinking since these materials will likely never be cooled again below the gelation temperature? Rheological measurements confirm that crosslinking at different temperatures results in up to a 5 times increase in shear storage modulus. The author's claim in several places that this change in mechanical properties arises even with no change in the gel structure, yet the SAXS data in Figure 3 clearly shows a variation in the contribution of polymer bundles and free polymers to the architecture. The authors even comment on page 6 that "On cooling... the SAXS results of the crosslinked gels indicate the presence of a fraction of single polymers...". This dissociation of single polymers from the bundles is a change in architecture, even if it may be insignificant to the suggested biological applications. In any case, the authors should clarify this point and the language in the results and discussion indicating that there is no change in network architecture to correctly describe the conclusions made from their SAXS data.

Additional comments are as follows:

- Loss moduli data should be added to the manuscript or the SI because it may be a useful quantity for the suggested applications. Moreover, it is important to report whether, and/or the extent to which, covalent crosslinking of the bundles may impact the loss modulus.
- Do these hydrogels creep when characterized above their gelation temperature before covalent crosslinking of the bundles? To what extent and at what rate do these materials relax stress? Presumably covalent crosslinking of the bundles will prevent creep and stress relaxation.
- Although the properties of the hydrogels are tunable based on crosslinking temperature, it appears that all of them become identical in mechanical response (Figure S2) if heated to 50 °C and then cooled. This is severely limiting in any case where T_{application} > T_{cl} since the properties will change to match the properties at the elevated temperature. Could the author's comment on why the properties change after crosslinking at T_{cl} for 1 h?
 - o Does the incomplete crosslinking reaction continue at higher temperature?
 - o If so could you design a system that can 'lock' in the mechanical properties so that future exposure to elevated temperatures does not alter the mechanical response?
- The cooling data from the SI (Figure S2) should be added to Figure 2d to show that all of the shear storage moduli become identical after heating.
- Figure S2 is missing the description for the formulation that is being presented.
- The conversion data is shown for 800 seconds, but a crosslinking time of 1 h is used for the hydrogels. Could the authors present the degree of conversion at 1h?
- Figure 2 caption is missing a word after mechanical.
- Figure 2f is missing the temperature of crosslinking.
- Pg 4, 3rd lists "NB". Can the authors italicize this to indicate more clearly that it is a latin acronym.

- Are there SEM images for the hydrogels crosslinked at different temperatures? This would help in determining if the change in architecture is insignificant.
- On Page 4 in the paragraph beginning with 'From 5 °C' it is difficult to understand what the authors mean by 'all samples'.
 - o In the same paragraph, the authors claim that the heating beyond T_{cl} irreversibly tightens the bundle. This is ambiguous as it does not specify whether the tightening is due to increased crosslinking from residual cross-linker or simply from the heating itself affecting the architecture.
- Page 4 last paragraph it is unclear what 'preservation of mechanical properties' means.
- It is difficult to identify the significance or weight of the single polymers vs. polymer bundles in the SAXS data.
 - o The authors mention that the contribution of scattering from single polymers is insignificant. Is there a quantitative method for using the SAXS fitting parameters to compare the contribution of scattering from the single polymers to the polymer bundles at elevated temperatures?
 - o It may be easier to interpret the decrease in contribution from single polymers in Figure 3d if the plots aren't shifted, similarly to Figure 3a.
- What are the absolute values of the storage modulus for the various crosslinking densities reported in Figure 4a?
- Figure 4b is not related to crosslink density as stated in figure title.

The reported approach is very interesting and provides an important design scheme that is likely to be appreciated by the readership of Nature Communications. The studies are generally well done and once my comments are addressed, especially regarding changes to the discussion of the results, I believe this work will be suitable for publication.

Reviewer #3 (Remarks to the Author):

This manuscript describes an interesting approach to generate hydrogels, using a secondary covalent crosslinking strategy to reinforce a physically crosslinked system. The topic is important to the hydrogel field, and likely to be of interest to researchers in that area. The authors provide substantial rheological and other characterization of the resulting hydrogels, which is a significant strength. The work is interesting, but the authors motivate this work by the importance of such systems for biological studies and for use in biological systems, including cell culture experiments. However, they do not provide any data from these types of studies. A paper focused purely on gel synthesis and characterization, even while quite interesting, is unlikely to be of interest to a broad audience. Publication in a materials-focused journal such as those in the Advanced Materials family or Polymer would likely be more appropriate. In addition, several key claims from the materials perspective require better experimental validation.

Specific concerns:

A key question with this system is whether the click crosslinking only occurs after polymer bundling, or if it happens to some extent before or concurrently. If it is before or concurrent, then it will lock in the gel architecture to some extent, and so will have an impact even though the claim is that this strategy allows a clean separation of mechanical properties from gel architecture. The authors should provide data, at the varying gelation T values, of a clean and quantitative temporal separation of the two modes of crosslinking.

The major claim of this manuscript, in terms of novelty, is that the architecture of the hydrogel networks is constant. The authors support this claim with some imaging, and analysis using SAXS. However, the imaging is qualitative, and differences in the networks (e.g., pore size) seem visible. The SAXS analysis utilizes a number of fit parameters, and the accuracy is unclear. From the data provided it is currently not possible to determine with any certainty whether network parameters are constant or varying in a statistically significant manner.

T is used as a major variable to create networks with varying mechanical properties, with the intended application being in the study of biological questions such as how cell behavior is impacted by mechanics. While the authors clearly obtain distinct and varying mechanical properties in the gels crosslinked at various temperatures (e.g., 25 to 50C), when they are subsequently heated the mechanical properties converge. This will significantly limit the range of mechanical properties over which studies can be done, as mammalian cell culture studies and biology more generally are performed at body T, which in humans is 37C. This will eliminate much of the differences between the hydrogels that are described in the manuscript. One could instead initially form the hydrogels at varying elevated T, and then cool to 37C, as demonstrated in the data provided, for cell culture studies. However, the range of mechanical properties the authors demonstrate with this approach is again very limited, with only a factor of approximately 2-fold change.

Reply to reviewer comments

Reviewer #1 (Remarks to the Author):

The authors create hydrogels consisting of bundled polyisocyanide (PIC) polymers by heating a solution above its LCST. The resulting bundled hydrogels are then fixated using difunctional crosslinkers connecting two azide groups along the polymer backbone. For the crosslinker, two different lengths are used; 3nm (roughly the inter-azide spacing distance along the PIC backbone) and 10 nm.

The manuscript argues that this chemistry provides control over mechanical properties in three ways:

- Crosslinking temperature
- Crosslinker design
- Crosslink density

Each of these three is quite convincingly demonstrated to have an effect on the mechanics.

I find these results interesting, novel, and sound. There is certainly sufficient evidence that each of the three control methods has some effect, but my main objection is that the manuscript does little more than note these effects. I find the mechanistic understanding of what is happening lacking, and would have very much liked to see the authors make more of an effort to present a coherent scenario for the changes to polymers, bundles, and network their interventions provoke, and how these lead to the observed changes in mechanical response.

Reply. We thank the reviewer for the kind words and we now more clearly presented the requested ‘coherent scenario’ in the first paragraph of the discussion of the manuscript.

The crosslinking scenario itself is clear enough; above LCST polymers are closer together in bundles giving particularly the shorter linker much more opportunity to bind intramolecularly. Much of these links survive cooling down again, preserving - to some extent - the network structure.

Two claims in the manuscript in particular deserve further motivation or explanation.

1. What does it mean that ‘crosslinking captures the architecture and the mechanics’? One of the principal reasons that networks stiffen is that not only the polymers, but particularly the bundles behave as stiff (or as these authors say semiflexible) chains. Surely, the additional intrabundle crosslinks change this effective stiffness profoundly in a largely non-temperature related way, so why would they approach a common asymptote? Clearly it cannot both be true that architecture and stiffness are preserved, because then the low-temperature parts would be the same too, right?
2. Why is the plateau modulus for the 2b linker so much lower? Ascribing it to the ‘loose bundle structure’ is insufficient. The plateau modulus depends, typically, on the persistence length, the concentration, and the mesh size - which of these is affected by the longer spacer and how? How does this compare to the bundle radius measurements in Fig 3f? These appear to show a relatively small effect? Could the Kholodenko model not also be used to measure a tentative change in persistence length?

Reply: Both questions are related, so we will address them together. The reviewer is right, of course one cannot keep the stiffness and the architecture constant and still have a large change in mechanical behaviour. In fibrous networks, indeed, the mechanics is strongly correlated to the persistence length of the bundles. Heating our PIC networks beyond the gelation temperature induces the formation of a bundled network and the temperature determines the persistence length of the constituent polymer chains and thus of the bundle¹. Higher temperatures give stiffer bundles (higher persistence length). As the interactions between the chains are strong in this regime, the addition of crosslinks does not impact

the mechanics (see Fig. 2a). Cooling below T_{gel} , however, removes the inter-chain interactions and now the network is solely held together by the crosslinks (maintaining the network architecture). Short crosslinkers allow for limited relaxation of the bundles and their mechanical properties closely represent those at the crosslinking conditions. Samples crosslinked at different temperatures then indeed have the same network architecture, but a different bundle persistence length $l_{\text{p,B}}$. Unfortunately, we cannot determine (fit) $l_{\text{p,B}}$ by SAXS (remark 2), since the angles are too small for our experimental setup in Grenoble. A much longer crosslinker (remark 2) on the other hand, does allow for structural relaxation of the bundles and in these materials, resulting in a much decreased persistence length, compared to the crosslinking conditions. In this case, the architecture does not change considerably. Experimentally (SAXS, Fig 3f), we do not see much change in the structure or the dimensions of the bundles, most likely because the bundles are not very well-defined in the first place (unlike some biological polymer for example).

Clearly, this explanation was not evident from our manuscript and we addressed it by rewriting the section under Figure 2, discussing the relation between the (crosslinked) network and the mechanical properties. In addition, we phrased the section of the crosslinker length and the loose-bundle structure more carefully, along the lines discussed above. In addition, we added the fitting results from the scattering experiments to the Supplementary Information (Supplementary Table 2).

3. The architecture may well be conserved, but I suspect the crosslinking induces and possibly locks in stresses in the gel. These stresses may to offset the mechanical response in ways not considered here at all (but potentially responsible for some of the effects, particularly the higher-T stiffening). How have the authors ruled out that this is the case?

Reply: This is a very interesting suggestion by the reviewer and, in fact, we did actively pursue locking-in strain which would lead to fantastic mechanics in very soft materials. We tried to do this by applying a large stress during the crosslinking process, but unfortunately, we saw no significant effects when the stress was removed. To be honest, we cannot exclude that some stress is built-in during crosslinking, but we can say that even when some stress is built-in, the mechanical consequences are small.

For completeness, we included the results of these experiments in the supporting information (Supplementary Fig. 7).

4. Overall, I found the manuscript interesting but somewhat disjointed and enumerative. Potentially, this may be repaired with more mechanistic or model support.

Reply: We addressed these comments by taking the different approaches outlined in the manuscript together and present a clear picture of the different effects (first paragraph of the discussion). A good model of these heterogeneous, bundled and crosslinked polymers would be fantastic, but the theoreticians that we work with are not ready yet.

Reviewer #2 (Remarks to the Author):

The manuscript by Schoenmakers et al. describes a hydrogel material system with temperature-dependent crosslinking that is used to tune the mechanical properties. It is shown that by controlling the temperature of crosslinking, the mechanical properties are altered without changing the architecture of the network. It is suggested that an independent control over the shear storage modulus and network architecture is desired

for biological applications. SAXS is used to find the bundle diameter of the crosslinked network and it is reported that the architecture of the gel is composed of both polymer bundles and single polymers. The approach for secondary covalent crosslinking of the hydrogel within the bundles (intra-bundle) and with an insignificant amount of intramolecular crosslinking is laid out well and may serve as an intelligent design scheme for future work. My main issue is with the potential utility of these materials. Upon gelation at elevated temperatures, covalent intrabundle crosslinking is used to "lock-in" the structure, allowing one to cool the system and retain the mechanical properties and structure. In a biological system where T_{application} is already above the gelation temperature, is much benefit to the covalent crosslinking since these materials will likely never be cooled again below the gelation temperature? Rheological measurements confirm that crosslinking at different temperatures results in up to a 5 times increase in shear storage modulus. The author's claim in several places that this change in mechanical properties arises even with no change in the gel structure, yet the SAXS data in Figure 3 clearly shows a variation in the contribution of polymer bundles and free polymers to the architecture. The authors even comment on page 6 that "On cooling... the SAXS results of the crosslinked gels indicate the presence of a fraction of single polymers...". This dissociation of single polymers from the bundles is a change in architecture, even if it may be insignificant to the suggested biological applications. In any case, the authors should clarify this point and the language in the results and discussion indicating that there is no change in network architecture to correctly describe the conclusions made from their SAXS data.

Reply: We are happy to read the positive comments of the reviewer regarding the concepts of our manuscript and are very grateful for the detailed comments below. The two main points that are raised: (a) Can you use this strategy for biological applications, considering the temperatures; and (b) is the architecture really constant?

- a. As for the temperatures, we presented the data for $T_{\text{crosslinking}} = 50\text{ }^{\circ}\text{C}$ and cooled down to $T = 5\text{ }^{\circ}\text{C}$ for the biggest effect. The effects at $37\text{ }^{\circ}\text{C}$ of course are still there, but obviously smaller. Although we would like to think of the work as more conceptual, we did address the challenge to make a bigger impact at $37\text{ }^{\circ}\text{C}$, which then requires higher crosslinking temperatures. We included an extra experiment with $T_{\text{cl}} = 65\text{ }^{\circ}\text{C}$, where indeed, the effect is more significant. To make a larger difference between crosslinked and non-crosslinked gels at $37\text{ }^{\circ}\text{C}$, one needs to raise the gelation temperature of the gel².
- b. Additionally, the reviewer is correct that the dissociation of bundles into single polymers is a change in architecture. The SAXS fitting data tells us that when crosslinked at $25\text{ }^{\circ}\text{C}$, a significant fraction of the polymers is not in the bundle (estimation 25%), This fraction grows when the sample is cooled to $5\text{ }^{\circ}\text{C}$ and disappears on heating to $50\text{ }^{\circ}\text{C}$. Sample crosslinked at $40\text{ }^{\circ}\text{C}$ or higher temperatures show less than 5 % contribution of single polymers irrespective of the measurement temperature.

We have clarified this analysis in the manuscript and phrased it more correctly. The fitting data has been added to the SI, Supplementary Table 2.

Additional comments are as follows:

- Loss moduli data should be added to the manuscript or the SI because it may be a useful quantity for the suggested applications. Moreover, it is important to report whether, and/or the extent to which, covalent crosslinking of the bundles may impacts the loss modulus.

Reply: This is a good suggestion as we sometimes focus too much on the elastic response of the gels, which is highly dominant and the loss data is very noisy. We added a plot of the loss modulus of a crosslinking experiment to the supporting information as well as a table with loss moduli data at T_{cl} and $T = 5\text{ }^{\circ}\text{C}$ (Supplementary Fig. 3).

- Do these hydrogels creep when characterized above their gelation temperature before covalent crosslinking of the bundles? To what extent and at what rate do these materials relax stress? Presumably covalent crosslinking of the bundles will prevent creep and stress relaxation.

Reply: Relaxation in the PIC gels, even in the absence of crosslinks is very slow, which allows us to use the pre-stress protocol for the nonlinear measurements. Indeed, crosslinking will further slow it down. We now mention this in the manuscript (p.3).

- Although the properties of the hydrogels are tunable based on crosslinking temperature, it appears that all of them become identical in mechanical response (Figure S2) if heated to 50 °C and then cooled. This is severely limiting in any case where $T_{\text{application}} > T_{\text{cl}}$ since the properties will change to match the properties at the elevated temperature. Could the author's comment on why the properties change after crosslinking at T_{cl} for 1 h?

Reply: The reviewer hits the nail on the head; to fully employ the possibilities of our approach, one needs to crosslink at higher temperatures than the application temperature. To demonstrate the potential at 37 °C, we included the data of the gel, crosslinked at 65 °C. To further clarify the potential at lower temperatures, we plotted the storage modulus at 20 °C of gels with different crosslinking temperatures.

- Does the incomplete crosslinking reaction continue at higher temperature?

Reply: We assume so, since the measured conversion is below completion. Changes in the bundle structure, induced by heating (or stress) may bring reactants in closer contact and induce further crosslinking. Experimentally, this is challenging to measure due to the very low concentration of reactive groups at this stage.

- If so could you design a system that can 'lock' in the mechanical properties so that future exposure to elevated temperatures does not alter the mechanical response?

Reply: This is an intriguing suggestion, but maybe challenging to achieve experimentally. It would require a scheme that releases or adds a component blocking the azides from further reaction, but only after the gel has been crosslinked. So far, we have not explored this interesting route. The stiffening process itself results from the entropic desolvation of the polymer and cannot be blocked for PICs.

- The cooling data from the SI (Figure S2) should be added to Figure 2d to show that all of the shear storage moduli become identical after heating.

- Figure S2 is missing the description for the formulation that is being presented.

Reply: This is a good suggestion and we moved the figure into the composite Figure 2, where it became panel f. We updated the caption and included the formulation.

- The conversion data is shown for 800 seconds, but a crosslinking time of 1 h is used for the hydrogels. Could the authors present the degree of conversion at 1h?

Reply: Already after a few minutes, most conversion plots show a plateau. To equilibrate the samples properly and to avoid generating samples with different thermal history, we set the crosslinking time for all samples to an excessive 1 h. Beyond 800 s, we did not collect any conversion data.

- Figure 2 caption is missing a word after mechanical.
Reply: This has been corrected.

- Figure 2f is missing the temperature of crosslinking.
Reply: This has been corrected.

- Pg 4, 3rd lists "NB". Can the authors italicize this to indicate more clearly that it is a latin acronym.
Reply: This has been corrected.

- Are there SEM images for the hydrogels crosslinked at different temperatures? This would help in determining if the change in architecture is insignificant.
Reply: The samples for SEM were crosslinked at 37 °C. Quantitative analysis of the images is virtually impossible, since we cannot control the sample thickness or water evaporation during sample preparation. Both affect the observed network density. In this case, SAXS as an in-situ technique is much more reliable.

- On Page 4 in the paragraph beginning with ‘From 5 °C’ it is difficult to understand what the authors mean by ‘all samples’.
- In the same paragraph, the authors claim that the heating beyond T_{cl} irreversibly tightens the bundle. This is ambiguous as it does not specify whether the tightening is due to increased crosslinking from residual cross-linker or simply from the heating itself affecting the architecture.
Reply (both remarks): ‘all samples’ refers to all the samples that were polymerized at different temperatures. Second remark: As the temperature increases, the persistence length of the polymer chain and thus the stiffness of the gel increases. Over the observed temperature regime, we find an exponential increase: $G' \sim RTe^{2\beta T}$ with $\beta = 0.055 \text{ K}^{-1}$ (Fig. 2a). After crosslinking, we observe the same (polymer chain) stiffening response (with $\beta = 0.051 \text{ K}^{-1}$). The process, we call tightening in the manuscript, but perhaps stiffening would be a better term. Interestingly, in the crosslinked samples, the effect is irreversible: after cooling the stiffness does not return to its original value, like we observe for the non-crosslinked gels. As the reviewer suggests, residual crosslinking that occurs after polymer chain reorganisation as a result of heating may very well be the origin. We rephrased the entire paragraph to elucidate this point.

- Page 4 last paragraph it is unclear what ‘preservation of mechanical properties’ means.
Reply: We rephrased this sentence (and much of the paragraph). See also reply to comment 2 of reviewer 1.

- It is difficult to identify the significance or weight of the single polymers vs. polymer bundles in the SAXS data? The authors mention that the contribution of scattering from single polymers is insignificant. Is there a quantitative method for using the SAXS fitting parameters to compare the contribution of scattering from the single polymers to the polymer bundles at elevated temperatures?
Reply: This is a relevant point. We do obtain relative weights from the SAXS curve fitting procedure, but this pre-factor contains both the volume fraction of the polymer and the scattering contrast. We now added a table containing the scattering fitting parameters to the Supplementary Information (Supplementary Table 2) and remarked on the contribution of the single polymers in the main text.

- It may be easier to interpret the decrease in contribution from single polymers in Figure 3d if the plots aren't shifted, similarly to Figure 3a.

Reply: We did shift the curves for clarity, but of course, one loses the option of direct mutual comparison. We added the un-shifted spectra to the Supporting Information and referred to them in the manuscript (Supplementary Fig. 5).

- What are the absolute values of the storage modulus for the various crosslinking densities reported in Figure 4a?

Reply: We have added a table of the absolute storage moduli of the materials with different crosslink densities to the Supporting Information (Supplementary Table 1). We prefer to display and discuss relative data, because of the effect of the hydrophobic DBCO group on the mechanical properties of the gel: A higher DBCO density increases the gel stiffness, even when they do not act as crosslinkers.

- Figure 4b is not related to crosslink density as stated in figure title.

Reply: The reviewer is correct and therefore we have altered the figure title.

- The reported approach is very interesting and provides an important design scheme that is likely to be appreciated by the readership of Nature Communications. The studies are generally well done and once my comments are addressed, especially regarding changes to the discussion of the results, I believe this work will be suitable for publication.

Reply: We acknowledge the assessment of the reviewer and hope that we addressed and applied all the comments satisfactorily.

Reviewer #3 (Remarks to the Author):

This manuscript describes an interesting approach to generate hydrogels, using a secondary covalent crosslinking strategy to reinforce a physically crosslinked system. The topic is important to the hydrogel field, and likely to be of interest to researchers in that area. The authors provide substantial rheological and other characterization of the resulting hydrogels, which is a significant strength. The work is interesting, but the authors motivate this work by the importance of such systems for biological studies and for use in biological systems, including cell culture experiments. However, they do not provide any data from these types of studies. A paper focused purely on gel synthesis and characterization, even while quite interesting, is unlikely to be of interest to a broad audience. Publication in a materials-focused journal such as those in the Advanced Materials family or Polymer would likely be more appropriate. In addition, several key claims from the materials perspective require better experimental validation.

Reply: We thank the reviewer for the assessment, but we disagree with him/her with the statement that the manuscript is solely of interest to materials and polymer scientists. For the first time, we have shown how low crosslink densities are able to stabilize networks and impact the mechanical properties. We already use the results in our biomedical work, as it is extremely easy to implement. Moreover, this work will certainly act as an inspiration to the biophysical community who tries to model realistic biological networks.

Specific concerns:

1. A key question with this system is whether the click crosslinking only occurs after polymer bundling, or if it happens to some extent before or concurrently. If it is before or concurrent, then it will lock in

the gel architecture to some extent, and so will have an impact even though the claim is that this strategy allows a clean separation of mechanical properties from gel architecture. The authors should provide data, at the varying gelation T values, of a clean and quantitative temporal separation of the two modes of crosslinking.

Reply: The reviewer is correct that it is important to investigate if the click reaction only occurs after polymer bundling, or that it also happens before or during the bundling process. To investigate this, we have carried out 2 additional experiments. In one experiment, we simply left the solution of network components at 5 °C (no bundling) for 2 hours whilst measuring the mechanical properties. For both crosslinkers **2a** (short) and **2b** (long), a very weak network is slowly formed, and after 2 hours the stiffness remains 2 orders of magnitude lower than that of the gels crosslinked in the gel. To exclude the effect of the lower temperature, we added 1 M of sodium iodide to increase the T_{gel} of the polymers to ~25 °C. Then, we crosslinked the solution at 18 °C, again in the absence of polymer bundles. Also, in this case, a very weak network is slowly formed. From these experiments, we conclude that the presence of the polymer bundles is crucial to obtain a crosslinked network, and that the crosslinking reaction at low T is negligible. The results of the described measurements have been added to the Figure 2 (panel c) of the main text (for the measurement at 5 °C) and as a Supporting Figure (NaI experiment) and we discussed them in the manuscript.

2. The major claim of this manuscript, in terms of novelty, is that the architecture of the hydrogels networks is constant. The authors support this claim with some imaging, and analysis using SAXS. However, the imaging is qualitative, and differences in the networks (e.g., pore size) seem visible. The SAXS analysis utilizes a number of fit parameters, and the accuracy is unclear. From the data provided it is currently not possible to determine with any certainty whether network parameters are constant or varying in a statistically significant manner.

Reply: Indeed, it is justified to say that it is challenging to analyse the network architecture quantitatively. Despite the fitting parameters, that we mostly obtained from blank experiments, the SAXS measurements provide the most reliable parameters. In the Supplementary Information, we provide an overview how we carry out the SAXS data analysis and the fitting results are given in Supplementary Table 2.

From the SEM images can be determined that the network is very heterogeneous, but a similar overall network structure and pore size is observed in images 1a-c. One needs to bear in mind that in these SEM pictures several layers of network are observed, which leads to an optically smaller pore size than is actually the case.

3. T is used as a major variable to create networks with varying mechanical properties, with the intended application being in the study of biological questions such as how cell behavior is impacted by mechanics. While the authors clearly obtain distinct and varying mechanical properties in the gels crosslinked at various temperatures (e.g., 25 to 50C), when they are subsequently heated the mechanical properties converge. This will significantly limit the range of mechanical properties over which studies can be done, as mammalian cell culture studies and biology more generally are performed at body T , which in humans is 37C. This will eliminate much of the differences between the hydrogels that are described in the manuscript. One could instead initially form the hydrogels at varying elevated T , and then cool to 37C, as demonstrated in the data provided, for cell culture studies. However, the range of

mechanical properties the authors demonstrate with this approach is again very limited, with only a factor of approximately 2-fold change.

Reply: The reviewer, like reviewer 2 (remark 1), is correct that the difference in mechanical properties of the gels at 5 °C is more evident than at higher temperatures. To highlight the applicability in biomedical applications, we added a crosslinking experiment at higher temperatures ($T_{cr} = 65$ °C).

References

- 1 Kouwer, P. H. J. *et al.* Responsive biomimetic networks from polyisocyanopeptide hydrogels. *Nature* **493** (2013) 651.
- 2 Kouwer, P. H. J. *et al.* Controlling the gelation temperature of biomimetic polyisocyanides. *Chin. Chem. Lett.* **29** (2018) 281.

Reviewers' comments:

Reviewer #1 (Remarks to the Author):

The authors have addressed most of my initial concerns, and in those cases where this wasn't possible - either because the mechanism was unclear or because the experiments did not warrant a firm statement - have been upfront. The additional explanations, particularly of the distinction between the various crosslinkers, is helpful (at least to me), as are the changes to the discussion.

I remain somewhat puzzled by the nature of the bundled state, which in some places is invoked to help explain the observed behavior, but in others (SAXS interpretation) are apparently poorly defined and should not be taken too literally as bundles similar to those seen in hierarchical biomaterials.

This issue will have to be resolved in follow up; even without this precise picture the approach and results in this paper are interesting and sound and likely to appeal to a broad readership. I am in favor of publication.

Reviewer #2 (Remarks to the Author):

The authors have addressed all of my questions and concerns. Specifically, they clarified the language in the manuscript to more accurately convey their results and address its limitations. I recommend for publication.

Reviewer #3 (Remarks to the Author):

In response to the earlier suggestion that including biological studies would add significantly to the interest in this work by a broad readership, the authors declined to include new data but indicated instead "We already use the results in our biomedical work, as it is extremely easy to implement." If they have already generated this type of data, and it is easy to do, they should include in the current manuscript as it would make the manuscript much more impactful.

As noted in the original review, to support the major claim that the architecture of the hydrogels was constant, the authors use analysis that are qualitative (SEM), and whose accuracy is unclear (SAXS). The authors acknowledge the limitations of these analysis in the response, and perhaps I am not seeing the changes, but I cannot find in the manuscript where these limitations are discussed, nor any tempering of the claims due to these limitations.

Minor comment:

In abstract, the authors state "...capturing the fibrous architecture exactly at the point of crosslinking" However, as the data does not indicate where specifically the covalent crosslinking occurs, but rather a range of distances, one cannot make this claim which suggests both types of crosslinking occur at the same molecular location.

Reply to reviewer comments (2)

Reviewer #1 (Remarks to the Author):

The authors have addressed most of my initial concerns, and in those cases where this wasn't possible - either because the mechanism was unclear or because the experiments did not warrant a firm statement - have been upfront. The additional explanations, particularly of the distinction between the various crosslinkers, is helpful (at least to me), as are the changes to the discussion.

I remain somewhat puzzled by the nature of the bundled state, which in some places is invoked to help explain the observed behavior, but in others (SAXS interpretation) are apparently poorly defined and should not be taken too literally as bundles similar to those seen in hierarchical biomaterials. This issue will have to be resolved in follow up; even without this precise picture the approach and results in this paper are interesting and sound and likely to appeal to a broad readership. I am in favor of publication.

Reply: The nature of the bundled state remains difficult to characterise. One of the key problems is the broad distribution present in such polymer networks: polymer length and bundle diameters are relatively broadly distributed averages. One should note that these bundle distributions are also present in biogels (sometimes even broader), but the order inside the bundles can be considerably higher, which makes analysis easier. Despite the poor characterisation tools, the bundle diameter is a crucial parameter in the mechanical properties of the materials and the reviewer is certainly right when he suggests that this will need future attention!

To emphasise this aspect in the manuscript, we changed the second line of the results from “*An aqueous PIC solution reversibly gels when heated above its lower critical solution temperature (LCST) and a branched, bundled gel is formed.*” into “An aqueous PIC solution reversibly gels when heated above its lower critical solution temperature (LCST) and a branched, bundled gel is formed with an architecture that is broadly distributed in length scales, both in bundle diameters and in pore dimensions.”

We thank the author for the comments and the recommendation.

Reviewer #2 (Remarks to the Author):

The authors have addressed all of my questions and concerns. Specifically, they clarified the language in the manuscript to more accurately convey their results and address its limitations. I recommend for publication.

Reply: we thank the reviewer for the comments and the recommendation to publish.

Reviewer #3 (Remarks to the Author):

In response to the earlier suggestion that including biological studies would add significantly to the interest in this work by a broad readership, the authors declined to include new data but indicated instead "We already use the results in our biomedical work, as it is extremely easy to implement." If they have already generated this type of data, and it is easy to do, they should include in the current manuscript as it would make the manuscript much more impactful.

Reply: Yes, we do currently use the technique in our biological work, but these studies are in progress and it will not benefit anyone to add some unchecked preliminary biological data to a conceptual manuscript like this. We request some additional patience from the reviewer and hope to present the biological results soon.

As noted in the original review, to support the major claim that the architecture of the hydrogels was constant, the authors use analysis that are qualitative (SEM), and whose accuracy is unclear (SAXS). The authors acknowledge the limitations of these analysis in the response, and perhaps I am not seeing the changes, but I cannot find in the manuscript where these limitations are discussed, nor any tempering of the claims due to these limitations.

Reply: This is a good suggestion to add to the manuscript. For the cryoSEM, we added to the caption of Figure 1 the sentence: “The cryoSEM images should be used for qualitative interpretation only, since the thickness of the sample strongly influences the apparent pore size of the gel.” For the SAXS, we state that [the network architecture] “is challenging to study in-situ, due to the low polymer concentrations and the large distributions in bundle and pore dimensions” and we added a line to the Methods section: “The polymer bundle radius R_B contains a normal distribution that represents inhomogeneity of bundle diameters”.

Minor comment:

In abstract, the authors state "...capturing the fibrous architecture exactly at the point of crosslinking" However, as the data does not indicate where specifically the covalent crosslinking occurs, but rather a range of distances, one cannot make this claim which suggests both types of crosslinking occur at the same molecular location.

Reply: We rephrased the abstract following the suggestion of the reviewer to clarify the concept. We thanks the reviewer for the comments that improved the manuscript.